# The Promotional Effect of GW4869 on *C. albicans* Invasion and Cellular Damage in a Murine Model of Oral Candidiasis

**DOI:** 10.3390/pathogens11121522

**Published:** 2022-12-12

**Authors:** Miaomiao Zhang, Ruowei Li, Yifan Zhou, Ruiqi Xie, Jingjing Ma, Hong Liu, Yao Qin, Maomao Zhao, Ning Duan, Pei Ye, Wenmei Wang, Xiang Wang

**Affiliations:** Nanjing Stomatological Hospital, Medical School of Nanjing University, Nanjing 210008, China

**Keywords:** *C. albicans*, GW4869, oral candidiasis, murine model, ultrastructure

## Abstract

*Candida albicans* (*C. albicans*) is one of the most common fungi in the human body; it is an opportunistic pathogen and can cause candidiasis. Extracellular vesicles (EVs) derived from the host cells have a potentially protective effect against pathogens and can be developed as vaccine formulations. GW4869 can inhibit the production and release of EVs. Previous studies have indicated that GW4869 can alter the immune and inflammatory responses of the host. However, the effect of GW4869 on *Candida* infection and the anti-*Candida* response of the host has not been investigated. We evaluated the effect of GW4869 on *C. albicans* invasion, biofilm formation, and cellular damage in a murine model of oral candidiasis. In this study, *C. albicans*-infected mice were injected with or without GW4869. The results proven by macroscopic, microscopic, and ultramicroscopic methods showed that GW4869 treatment exacerbated the oral candidiasis of mice, promoted *C. albicans* invasion and biofilm formation, and aggravated oral mucosal inflammation and cellular ultrastructural damage. The results are beneficial in the further exploration of the immune mechanism of *C. albicans* infection.

## 1. Introduction

*Candida albicans* (*C. albicans*) is one of the most common opportunistic pathogens in the human body, colonizing the mouth, skin, and the gastrointestinal and urogenital tracts [1,2,3]. *C. albicans* is harmless in healthy individuals; however, organism immunity or local environmental changes can make *C. albicans* exert its pathogenicity. *C. albicans* can result in life-threatening systemic infections with high mortality rates [4,5]. *Candida* infections are more severe in immunocompromised individuals and in patients with implanted medical devices [6]. *C. albicans* biofilms could be one of the reasons for antifungal treatment failure. Thus, the need for an updated understanding of the antifungal response of the host against oral candidiasis and *Candida* biofilms is urgent.

Extracellular vesicles (EVs) are released from widespread cell types and participate in cell-to-cell communication under physiological and pathological conditions [7,8,9]. GW4869 is available and inexpensive, and is able to inhibit the production and release of EVs. It has been shown that GW4869 has a regulatory effect on immune responses [10]. A previous study indicated that GW4869 could not inhibit polymorphonuclear neutrophils motility, but it did abrogate orienting chemotactic migration of human neutrophils [11]. Moreover, GW4869 has been proven to decrease Th2 cytokines and house dust mite allergen-induced airway inflammation [12]. To date, however, the influences of GW4869 on *Candida* infection and the anti-*Candida* response of the host have not been studied.

In this study, we investigated the effect of GW4869 on *C. albicans* infection in the murine model of oral candidiasis and evaluated its effect on the oral mucosal barrier, and defensive and inflammatory responses, through macroscopic, microscopic, and ultramicroscopic methods. Our results showed that GW4869 not only had a promotional effect on *C. albicans* invasion and biofilm formation, but also exacerbated mucosal inflammation and cell damage. These results suggest a protective role for EVs derived from the host cells against *C. albicans* infection and cellular damage.

## 2. Materials and Methods

### 2.1. Culture of Fungal Cells

*C. albicans* strain SC5314, acquired from the Department of Mycology, Institute of Dermatology, Chinese Academy of Medical Sciences, was used for this study. The *C. albicans* strain was cultured in yeast extract–peptone–dextrose (YPD) medium for shaking at 30 °C and 180 rpm.

### 2.2. Animals

All of the murine experiments were performed in accordance with the ethical guidelines for animal experimentation and were approved by the local Ethics Committee [Institutional Review Board Approval Number: 2015NL-001 (KS) and 2018NL-008(KS)]. Female C57BL/6J mice (six-to-eight weeks) were purchased from Sino-British SIPPR/BK Lab Animal Ltd. (Shanghai, China). The mice were acclimatized to the experimental facility for one week to relieve stress. Ten mice were randomly divided into two groups of five animals each: the *C. albicans* SC5314 infection group (the control group), and the *C. albicans* SC5314 infection and GW4869 injection group (the GW4869 group).

### 2.3. Oral Candidiasis Model and GW4869 Treatment

To cause mice immune suppression and avoid bacterial co-infections, mice received 100 mg/kg prednisolone (Xianju Pharmaceutical Co., Taizhou, China) subcutaneously and 0.8 g/L tetracycline (Solarbio Science and Technology Co., Beijing, China) was added to the drinking water 24 h before *C. albicans* SC5314 infection. Then, the mice were anesthetized by injection with 1% pentobarbital sodium intraperitoneally. We placed a swab dipped in *C. albicans* SC5314 (1 × 10^7^ yeast/mL) at the base of the tongue for 2 h. The mice in the control group and the GW4869 group were submucosally injected with 20 µL 5% dimethyl sulfoxide (DMSO) and 20 µL solution of GW4869 dissolved in DMSO (2.5 µg/g) in white pseudomembrane lesions of the tongue, respectively. The mice were injected once per day for five days. After five injections, the mice in both groups were scheduled for death and their tongues were collected. This experimental process is shown in Figure 1.

### 2.4. Macroscopic Scoring of Candidiasis Lesions

Observers blinded to the experimental grouping evaluated the area of white lesions of murine tongues by using Image Pro Plus Software (version 6.0), as previously described [13]. Then, the significant difference between the control and GW4869 groups was analyzed by comparing the area of both groups.

### 2.5. Histopathologic Examination and Periodic Acid Solution and Schiff Staining

All tissues of the murine tongue were fixed for 24 h in 4% paraformaldehyde, and were then made into paraffin blocks. We cut the paraffin blocks into 3mmthick sections and stained them with hematoxylin and eosin (H&E) and periodic acid solution and Schiff reagent (PAS). The experiment was conducted by two blinded observers who analyzed the severity of the inflammation according to the degree of infiltration of the inflammatory cells. The inflammation degree of tongue tissues of each mouse was evaluated based on 10 images randomly taken at 200×magnification of H&E staining, and the average value represented the inflammation degree of this mouse. The scale from 0 to 3 was as follows: 0, five or fewer inflammatory cells found in each high-power field; 1, a small number of inflammatory cells, surface mucosal epithelium only, less than one-third of the epithelium; 2, a moderate amount of inflammation, epithelium not to exceed two-thirds; and 3, a large number of inflammatory cells, occupying the whole epithelium [14]. The observers blinded to experimental grouping evaluated the extent to which *C. albicans* had invaded on a scale from 0 to 4, according to the PAS staining. The invasion degree of *C. albicans* of each mouse was evaluated based on 10 images randomly taken at 200×magnification of PAS staining, and the average value represented the invasion degree of this mouse. The scale was as follows: 0, none; 1, the invasive range was between 1% and 25%; 2, the invasive range was between 26% and 50%; 3, the invasive range was between 51% and 75%; and 4, the invasive range was between 76% and 100% [15].

### 2.6. Scanning Electron Microscopy

Freshly isolated tongue tissues were rinsed and cut into 2 mm × 2 mm tissue pieces. Next, the samples were fixed in an electron microscope fixative of 4% at 4 °C for four hours. Ten minutes were spent soaking the samples in 0.1 mol/L phosphate buffer, which was repeated three times; they were then dehydrated with ethanol at a gradient of increased concentration and replaced with amyl acetate for 10 min. Finally, they were dried at critical point and sprayed gold. We observed and photographed the samples under scanning electron microscopy (SEM). The experiment was conducted by two observers who were blinded to the experimental grouping and evaluated the number of hyphae and spores of *C. albicans*. The hyphae and spores of each mouse were counted based on 10 images randomly taken at 1000× magnification by SEM, and the average value was considered as the representative value of this mouse.

### 2.7. Transmission Electron Microscopy

We processed the tongue tissue samples using transmission electron microscopy (TEM). After being gradient dehydrated with ethanol, they were replaced with epoxypropane for 10 min, and then soaked and encapsulated with epoxy resin. Then, we stained TEM slices from 50 to 70 nm after polymerization with uranium acetate and lead citrate. TEM was used to observe and photograph the treated slices. We calculated the size and average optical density (AOD) of the mitochondria and compared them between the control group and GW4869 group. The experiment was conducted by two observers who were blinded to the experimental grouping and evaluated the size and average optical density (AOD) of the mitochondria. The size of the mitochondria of each mouse was the mean of all of the maximum diameters of the mitochondria measured by Image Pro Plus based on 10 images randomly taken at 5000× magnification by TEM. The mitochondrial AOD of each mouse was the mean of all mitochondrial AOD measured by Image Pro Plus based on 10 images randomly taken at 5000× magnification by TEM.

### 2.8. Fluorescence

The tongue tissue samples were cut into 20µm thick slices. The slices were fixed with 4% paraformaldehyde for 4 h at 4 °C and then dehydrated with 30% sucrose solution for 24 h at 4 °C. After being soaked in phosphate-buffered saline (PBS) for 15 min, the slices were blocked with 3% bovine serum albumin (BSA) for 1 h at 37 °C. Then, we added the slices to the suspension of 5µL of Component B (Calcofluor white M2R) and then incubated the slices at 30 °C in the dark for 30 min. Finally, we observed and photographed the slices with confocal microscopy. Hyphae were stained by Calcoflour white M2R, which labeled cell-wall chitin with blue fluorescence [16]. The experiment was conducted by two observers who were blinded to the experimental grouping and evaluated the fluorescence intensity and cell wall thickness of *C. albicans* taken at 1000× magnification by confocal microscopy. The fluorescence intensity of each mouse was the mean of all of the fluorescence intensity measured by Image Pro Plus from 10 random images. For each mouse, 10 images were randomly selected and for each image, 20 *C. albicans* cells were randomly selected to measure the cell wall thickness by Image Pro Plus. Finally, the cell wall thickness for each mouse was an average value of 10 images.

### 2.9. Statistics

The mean ± SD is expressed for all of the results. In order to perform statistical analysis, we used GraphPad prism (version 7.0). The data were analyzed by independent study *t*-test; *, *p* < 0.05; **, *p* < 0.01; ***, *p* < 0.001.

## 3. Results

### 3.1. GW4869 Promoted Oral Candidiasis and Biofilm Formation

Through macroscopic observation, we found that the white lesions on the tongues of the *C. albicans* + GW4869 group were larger than those of the *C. albicans* group (Figure 2A). In comparison with the *C. albicans* group (0.23 ± 0.59 cm^2^), the area of lesions in the *C. albicans* + GW4869 group (0.37 ± 0.71 cm^2^) was significantly higher (*p* < 0.01) (Figure 2B). The results showed that GW4869 played a promotional role in *C. albicans* growth and biofilm formation.

### 3.2. GW4869 Aggravated C. albicans Infection-Induced Mucosal Inflammation

The result of the H&E staining indicated that a large number of inflammatory cells were distributed intensively throughout the whole mucosal epithelium in the *C. albicans* + GW4869 group; the inflammatory cells occupied two-thirds of the mucosal epithelium in the *C. albicans* group. After treatment with GW4869, more severe microbial infections could be seen in the epithelium (Figure 3A). The degree of inflammation of the *C. albicans* + GW4869 group (2.80 ± 0.45) was higher than that of the *C. albicans* group (1.80 ± 0.84) (Figure 3B). This result indicated that GW4869 could contribute to more inflammatory cells infiltrating the epithelium.

### 3.3. GW4869 Increased the Invasion of C. albicans into Mucosal Epithelium

We observed hyphae of *C. albicans* with PAS staining. Although hyphae occupied the majority of the epithelium in both groups, the density of hyphae in the epithelium of the *C. albicans* + GW4869 group was significantly higher than that of the *C. albicans* group (Figure 4A). Compared with the *C. albicans* group (2.00 ± 0.71), the *C. albicans* + GW4869 group (3.40 ± 0.89) had more hyphal invasion and colonization (Figure 4B). This result showed that GW4869 augmented the colonization and invasion of *C. albicans* in the oral epithelium.

### 3.4. GW4869 Facilitated C. albicans Growth and Bacterial Biofilm Formation

As shown in Figure 5A, we clearly observed extensive biofilms and keratin desquamation on the tongue mucosal surface under SEM. In addition to hyphae and spores of *C. albicans*, rod-shaped bacteria intensely distributed in the biofilms were common. Specifically, in the *C. albicans* group, the biofilm was smoother, but keratin desquamation and bacteria were less than in the *C. albicans* + GW4869 group. The *C. albicans +* GW4869 group had more hyphae (14.40 ± 3.51) than the *C. albicans* group (7.20 ± 1.30), and the same was true for the spores (19.20 ± 4.97 vs. 13.60 ± 1.14, respectively) (Figure 5B). Therefore, these results indicated that GW4869 could promote the proliferation of *C. albicans* and bacterial biofilm formation on the surface of the tongue.

### 3.5. GW4869 Exacerbated C. albicans Infection-Induced Ultrastructural Damages of Oral Mucosal Epithelial Cells

To assess the effect of GW4869 on *C. albicans* infection-induced oral mucosa alterations in both groups, we observed and analyzed the ultrastructure of the epithelial cells under TEM. Clearly, swelling of the mitochondria, shrinking of the nuclear envelope, and less endoplasmic reticulum were present in the *C. albicans* + GW4869 group (Figure 6A). Significant changes in the mitochondria occurred in the *C. albicans* + GW4869 group, which manifested as severe swelling of the mitochondria and disintegrating of the mitochondrial cristae. The group with GW4869 treatment (0.89 ± 0.93 μm) had alarger mitochondrial size than the group without GW4869 treatment (0.64 ± 0.57 μm) (Figure 6B), and it had a lower mitochondrial AOD (0.41 ± 0.02) than that without GW4869 treatment (0.62 ± 0.07) (Figure 6C). These data suggested that GW4869 could increase *C. albicans* infection-induced ultrastructural damage of the oral mucosal epithelial cells.

### 3.6. GW4869 Stabilized the Cell Wall Structure of C. albicans

GW4869 accentuated *C. albicans* proliferation and invasiveness in the oral mucosal epithelium. We used fluorescence staining to investigate the morphological alterations in the *C. albicans* hyphae in the oral mucosal epithelial tissues following GW4869 treatment. The group with GW4869 treatment (41,965.96 ± 6285.64) had a higher fluorescence intensity than the group without GW4869 treatment (27,764.68 ± 3606.89) (Figure 7B). The group with GW4869 treatment (0.75 ± 0.06) had a thicker cell wall than the group without GW4869 treatment (0.52 ± 0.10) (Figure 7C). This result indicated that GW4869 treatment could increase the chitin in the cell wall of *C. albicans*.

## 4. Discussion

*C. albicans,* one of the main resident species in the oral cavity, plays a key role in the occurrence and development of some oral diseases, such as oral candidiasis, oral leukoplakia, and oral cancer [17]. The change in *Candida* from a commensal state to a pathogenic state is influenced by local and systemic factors [18]. *Candida* pathogenicity is closely related to hyphae and biofilms [19,20,21]. Therefore, hyphae and biofilms can be used to evaluate the pathogenicity of *Candida.* Although many in vitro studies have examined *C. albicans* infection, the cellular responses and alterations in oral candidiasis need to be further studied in vivo. The results of this study, which utilized a murine model, indicated that GW4869 promoted *C. albicans* invasion, biofilm formation, mucosal inflammation, and oral epithelial cell damage in oral candidiasis.

Our study adopted multiple methods to demonstrate the impact of GW4869 on *C. albicans* infection and mucosal tissue damage. Through these methods, including H&E, PAS, SEM, TEM, and fluorescence staining, we clearly observed the macroscopic, microscopic, and ultramicroscopic changes caused by oral candidiasis with and without GW4869 treatment. GW4869 can decrease the release of EVs and inhibit the proliferation of individual cells within the host phagocytes. The protein and RNA cargo of EVs are critical in triggering the rapid proliferation of the host macrophages [22]. The protective effect induced by EVs is primarily dependent on the activation of phagocytes, antigen-presenting cells, and upregulating cytokine levels [23,24].

In the present study, we performed histopathological and ultrastructural examinations of the tongue mucosa to explore specific changes inside the mucosal tissues. After treatment with GW4869, we observed more neutrophils aggregating into the epithelium and more micro-abscesses. A recent study indicated that GW4869 significantly reduced β-Glucan-stimulated neutrophil secretion of IL-1α through EVs [25]. GW4869 also may suppress immune responses of neutrophils and other immunocytes against *C. albicans*, making it easier for *C. albicans* to invade the oral mucosal epithelium and induce inflammation. In addition, oral epithelial cells released EVs to exert antifungal effects, and GW4869 could weaken their antifungal effects [26].

In this study, the alterations in the cellular ultrastructure assessed by TEM indicated that GW4869 could remarkably augment *C. albicans* infection-induced damage to the organelles. The most obvious change appeared in the mitochondria, which showed severe swelling and disintegration of the mitochondrial cristae. As energy-supplying and signaling organelles, mitochondria are necessary for innate and adaptive immune responses [27]. Mitochondrial dysfunction significantly triggers inflammation and prevents the repolarization of inflammatory macrophages [28,29,30]. Conversely, the transfer of mitochondria through microvesicles (EVs with the larger size) leads to the enhancement of mitochondrial function [31]. Therefore, GW4869 may promote *C. albicans* infection by damaging the mitochondria of the host cells and impairing immune activation.

In addition, through fluorescence staining, we observed increased chitin in the cell wall of *C. albicans* in the mucosal epithelium following submucosal injection with GW4869. As a polysaccharide, chitin is a main component and it plays an important role in fungal cell wall structure stability and the interaction between fungi and the environment [32]. Chitin promotes the replication of fungi and helps fungi resist environmental pressures [33,34,35]. Therefore, the increase in chitin in *C. albicans* might suggest that the production and release suppression of EVs from the host could contribute to the growth and function of *C. albicans*. Currently, however, the specific effects of GW4869 on bacterial and fungal infections remain unclear. Although we investigated the effect of GW4869 on *Candida* infection, further study is needed to explore the depth of this mechanism.

Collectively, we established a murine model of oral candidiasis and demonstrated that GW4869 promoted oral candidiasis and biofilm formation by accentuating *C. albicans* hyphal growth and invasion, aggravating oral mucosal inflammation, and exacerbating ultrastructural damage of oral mucosal epithelial cells. An in-depth understanding of the effects of GW4869 on the innate immune and microbial infection could clarify the underlying mechanism of infection immunity.

## Figures and Tables

**Figure 1 pathogens-11-01522-f001:**
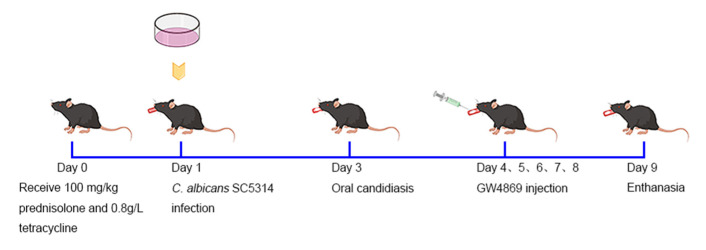
Experimental design is drawn using Figdraw. The time bar represents the strategy for immunosuppression, *Candida albicans* (*C. albicans*) infection, and GW4869 injection in the mice. Immunosuppression was given on Day 0 before *C. albicans* infection. Between Day 4 and Day 8, the mice were injected with GW4869 every day. On Day 9, the mice were euthanized and their tongues were collected.

**Figure 2 pathogens-11-01522-f002:**
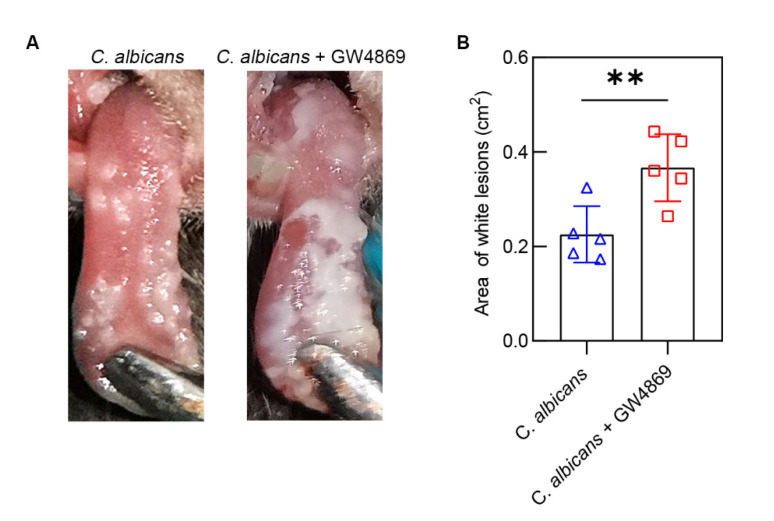
Macroscopic observation and area measurement of white lesions on the tongue. (**A**) White patches of tongue mucosal tissues. (**B**) Compared with the *C. albicans* group, the *C. albicans* + GW4869 group had a larger area of white lesions. Data are expressed as the mean ± SD. **, *p* < 0.01.

**Figure 3 pathogens-11-01522-f003:**
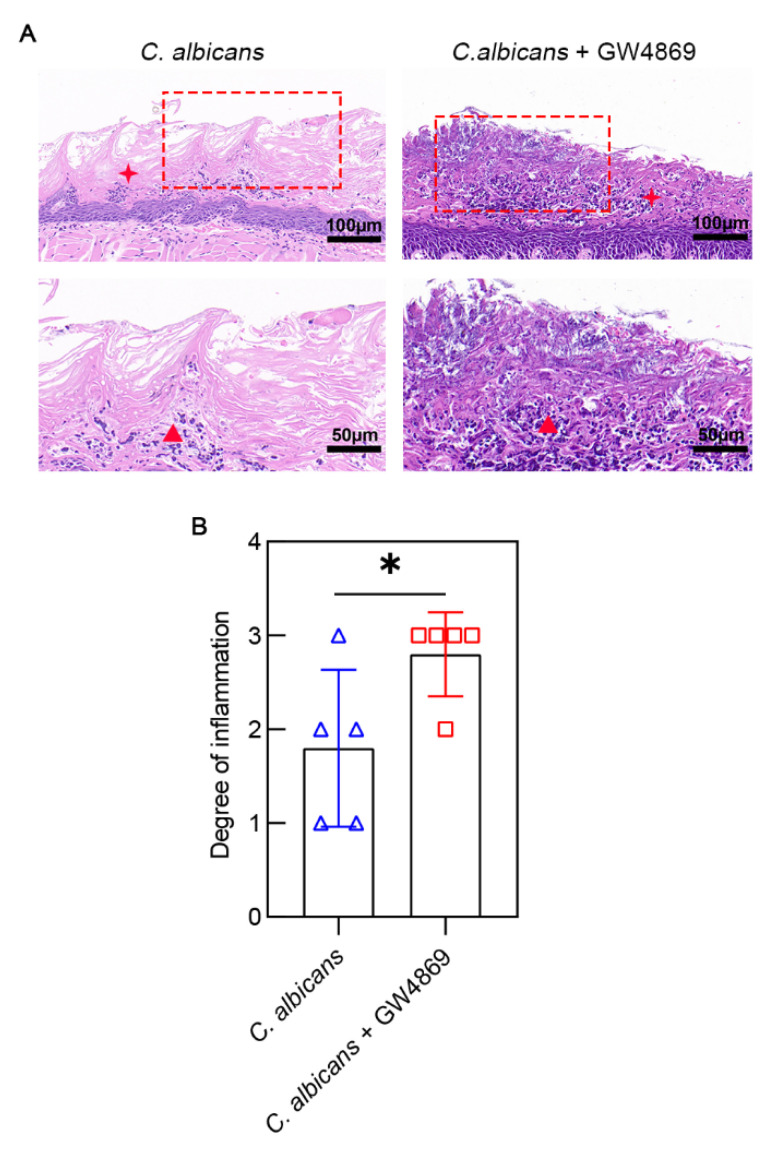
Histopathological changes of tongue tissues. (**A**) The lesions of tongue tissues were assessed by H&E staining. Thetongue tissues injected with GW4869 had more inflammatory cells than those not injected with GW4869. The images above have a scale of 100 µm. The images below have a scale of 50 µm. (**B**) There were different degrees of inflammation in both groups, and the group treated with GW4869 had a higher level of inflammation. Data are expressed as the mean ± SD. Inflammatory cells (▲), cornified layer (✦). *, *p* < 0.05.

**Figure 4 pathogens-11-01522-f004:**
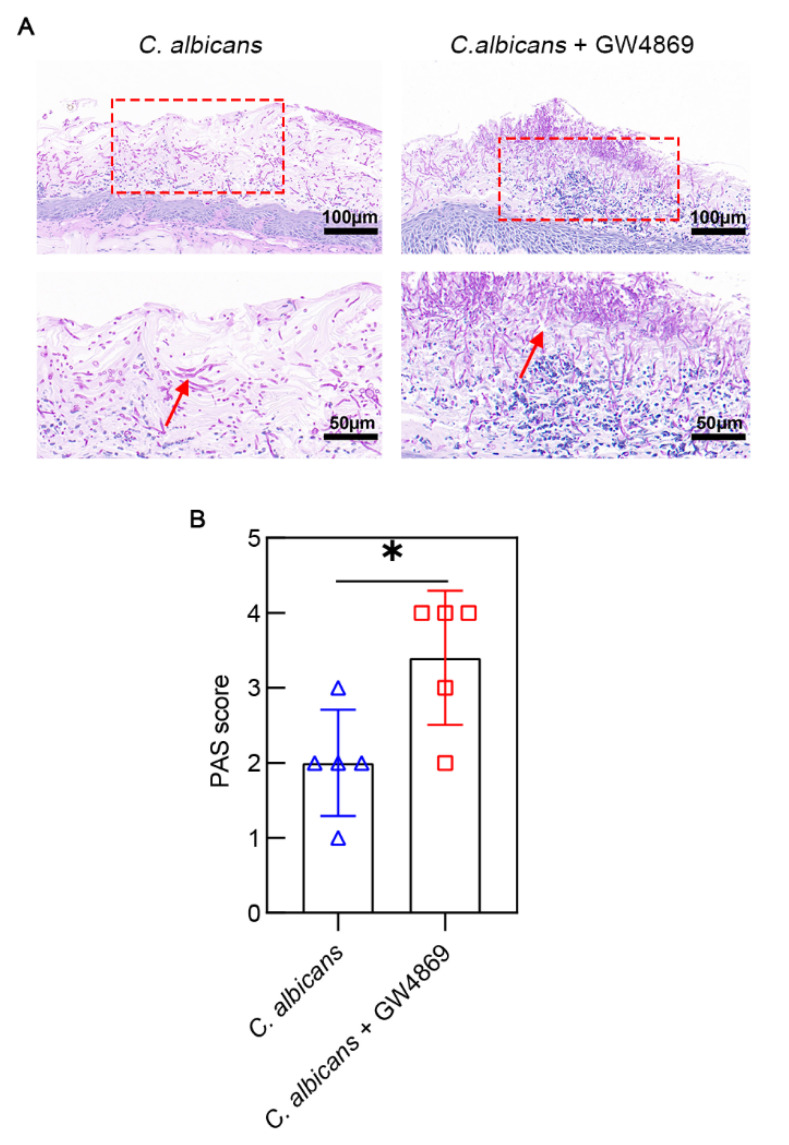
(**A**) There was higher density of hyphae in the group treated with GW4869. The images above have a scale of 100 µm. The images below have a scale of 50 µm. (**B**) The group treated with GW4869 had a higher PAS score. Data are expressed as the mean ± SD. Hyphae (↑). *, *p* < 0.05.

**Figure 5 pathogens-11-01522-f005:**
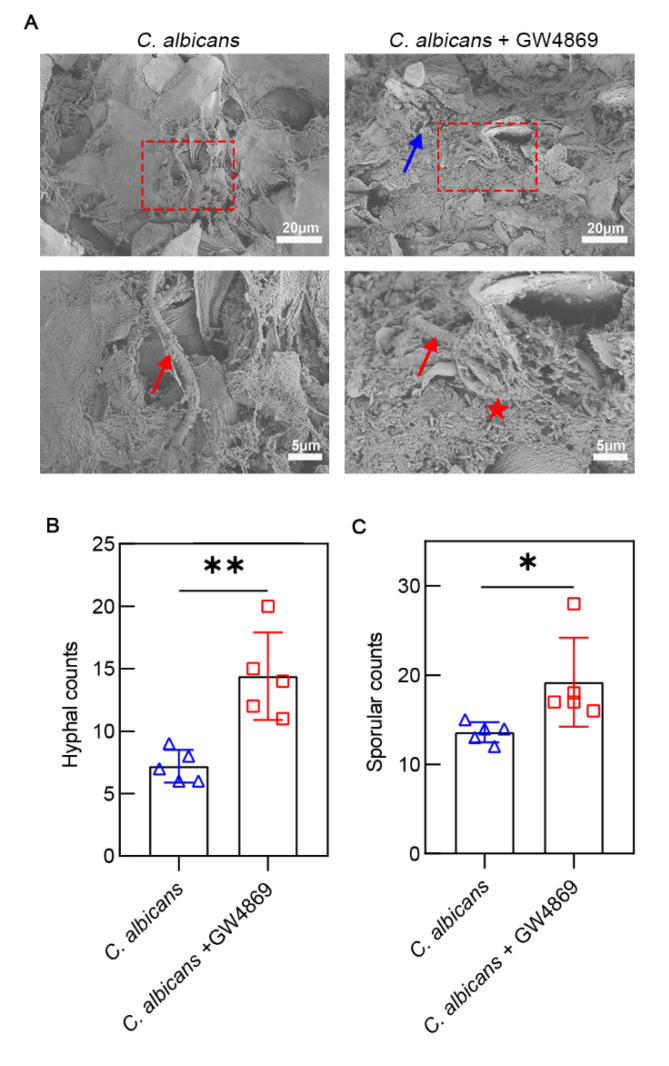
Scanning electron microscopy (SEM) evaluation of the murine tongue infected by *C. albicans.* (**A**) The group with GW4869 treatment had more bacteria than the group without GW4869 treatment. The images above have a scale of 20 µm. The images below have a scale of 5 µm. (**B**) The number of hyphae increased with GW4869 treatment. (**C**) The number of sporesincreased with GW4869 treatment. Data are expressed as the mean ± SD. Hyphae (↑); spores (↑); bacteria (★). *, *p* < 0.05; **, *p* < 0.01.

**Figure 6 pathogens-11-01522-f006:**
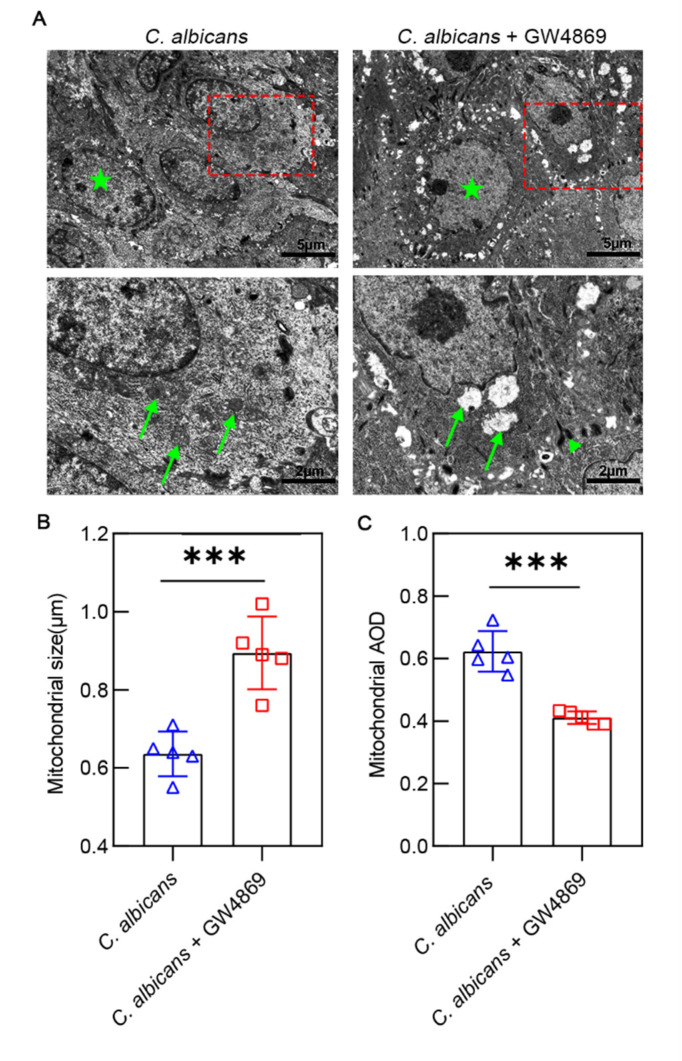
Cellular ultrastructure of the mucosal epithelium of the murine tongue under transmission electron microscopy (TEM). (**A**) In contrast with the *C. albicans* group, the mitochondria showed decreased intensity and severe swelling in the *C. albicans* + GW4869 group. The images above have a scale of 5 µm. The images below have a scale of 2 µm. (**B**) After being treated with GW4869, the mitochondrial size was larger. (**C**) After being treated with GW4869, the mitochondrial AOD was lower. Data are expressed as the mean ± SD. Cell nucleus (★), mitochondria (↑), desmosomes (▲). ***, *p* < 0.001.

**Figure 7 pathogens-11-01522-f007:**
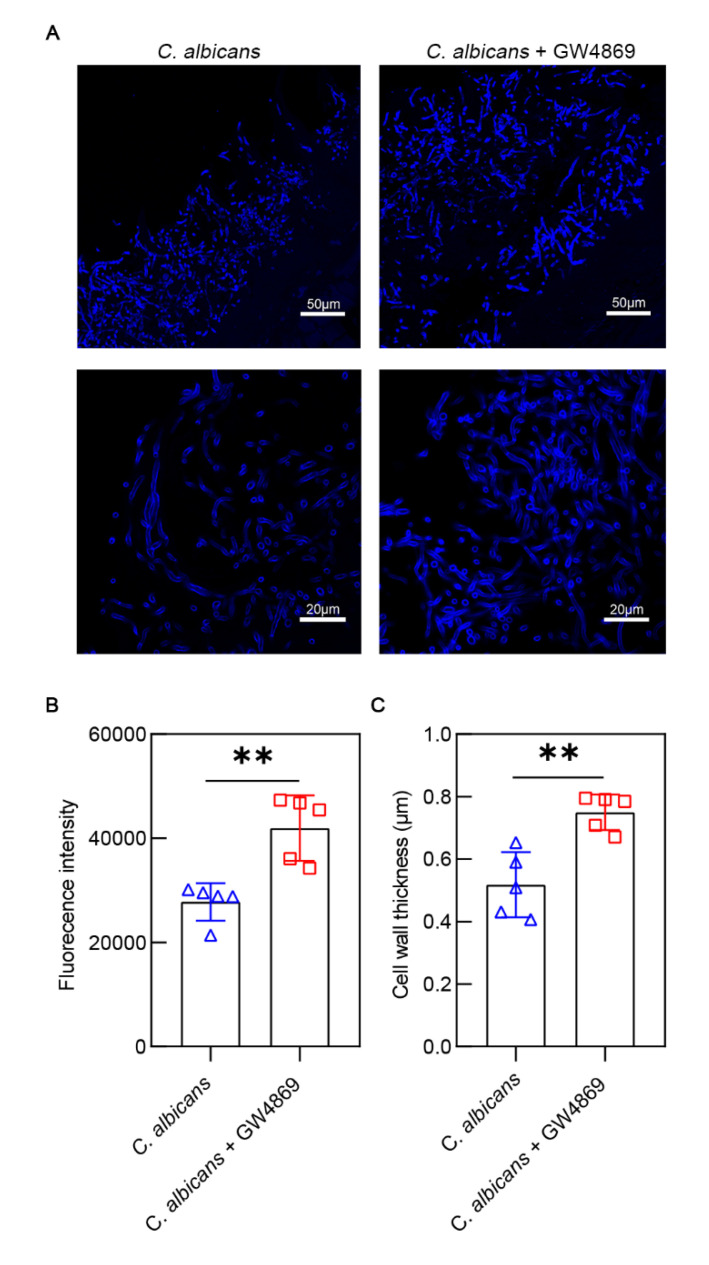
Fluorescence staining of tongue tissues infected by *C. albicans*. (**A**) Hyphae were stained by Calcoflour White M2R. The images above have a scale of 50 µm. The images below have a scale of 20 µm. (**B**) Hyphal fluorescence intensity was enhanced after GW4869 treatment. (**C**) The cell wall was thicker after GW4869 treatment. Data are expressed as the mean ± SD. **, *p* < 0.01.

## Data Availability

The data presented in this study are available upon request from the corresponding author.

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
