# Peer review of "The Promotional Effect of GW4869 on C. albicans Invasion and Cellular Damage in a Murine Model of Oral Candidiasis"

_pathogens, 2022, doi:10.3390/pathogens11121522_

Round 1

Reviewer 1 Report

The first thing is to congratulate you for the paper. It seems to me a very interesting article that describe into the role of EVs in Candida albicans infections, and how they minimize cell damage and the effects derived from the infection, which could serve in the future for the development of new therapies. Although I have recommended the publication of the article, I think that you should first take into account a series of recommendations that I make below, and that they can correct some important aspects in the article:

·   The bibliography section should be revised, since the numbering is wrong. There are two numbers 1, and 29 citations are referenced in the article. I think it's just the repeated number 1, since then the quotes correspond if one is subtracted from each of them.

·  In the abstract (line 13), it is stated that: “GW4869 is a neutral sphingomyelinase that inhabits the production and release of EVs”. Later, in the introduction (line 40) it is stated that: “GW4869, available and inexpensive, is a neutral sphingomyelinase inhibitor that inhibits the production and release of EVs”. It should be corrected in the abstract because it seems that they are different things.

·       The amount of sample, 5 control and 5 infected mice, seems very small. It would be good to clarify if previous statistical studies have been done to calculate the necessary sample size for this trial.

·     In section 2.3 of Materials&Methods (line 71-80), it should be clarified why mice are treated with prednisolone and tetracycline. It can be understood that it is to cause a state of immunosuppression and to avoid bacterial co-infections, but it would be good to clarify it in the text. In addition, it is also convenient to clarify why DMSO is administered together with GW4869 (adjuvant, transport...).

·    In Figure 1 it says: “Day 4, 5, 6, 7 Inject GW4869”. However, in the legend of the figure (line 84), it says “Between Day 4 and Day 8”.

·      Review some citations, since references to what is put in the text are not found in the citations of the bibliography:

o   In section 2.4 of Materials&Methods (line 87-90) a scale is commented that will be followed to evaluate the lesions produced by C. albicans in the tongue of mice, however the citations referred to do not reflect that scale. It would be convenient to include citations that have already established this criterion or explain why this classification has been followed.

o   In section 2.5 (line 97-101) reference is made to a citation that does not describe the scale that is being mentioned in the text, and the same thing happens in lines 101-105 and reference (14).

o   In section 2.8 of Materials&Methods (line 129-131) the citation refers to a method that is not described in the bibliography citation.

o   In Figure 6, the ER and the mitochondria indicated by the arrow cannot be distinguished very well. I don't know if they have any clearer or better resolution images.

·     In line 216 there is a wording error in the sentence, which is not well understood: “...oral mucosal epithelium ofWe used immunofluorescense...”.

Reviewer 2 Report

Introduction:  The intro and discussion need extensive editing for grammar.  In addition some of the claims made such as “C. albicans being the most common opportunistic pathogen” need qualifying statements or further citations.  Furthermore the EV discussion makes it sound like the EVs are coming from fungi as opposed to the human cells.  A more clear explanation is needed. GW4869 is never described outside of being a neutral sphingomyelinase inhibitor while this is a common drug a more substantial rational for its use would be informative.

Major Issues:

In most results 2 levels of magnification are given.  The rational for providing these two levels of magnification is unclear.  What images were used for quantitation is also unclear. Also in all graphs a bar graph is present but what the bar graph indicates in not clear, I assume average but it is not stated.  Also not stated what the thing that looks like an error bar is, I assume a standard deviation but it is not stated.  Uninfected mice were not used as a control in any experiments and GW4869 alone was not used as a control when looking at mitochondrial damage.  

Results: Figure 2:  Method of determining the amount of white lesion is not truly quantitative.  Figure 2A tongues look pretty similar. Also unclear what the bar is indicating in Figure 2B, same issue for 3 and 4 B as well.  

Figure 3+4:  How did the researchers decide where to take the image that they quantified?  It says that the whole morphology of the tongue was observed but it is unclear what this means.  How many sections were quantified from each animal?  Given the limited number of animals and replicates it is hard to determine if these results are significant.

Figure 5: 5A, authors claim there are more bacteria in the GW4869 group but it not possible to see these bacteria and no quantification is made.  5B unclear how hyphae were quantitated in 5C. same and 5B but also obvious single outlier, possibly not a significant difference when this is disregarded.

Figure 6 Interpretation of mitochondrial size is difficult to see.  Also claims of nuclear envelope and ER changes but no data or quantitation is given to support these claims.

Figure7 Authors claim hyphal intensity increased.  It is unclear how this was determined.  Also how cell wall thickness was measured is not stated. Authors also state this is immunoflour but then that is was stained with calcoflour which is not immunoflourescence.

Overall lack of details as to the quantitation of the images and lack of image clarity makes it difficult to assess the accuracy of the analysis.

Discussion:

Stated that C. albicans is one of the main resident species of the oral cavity,   it is unclear what this means and no citation is given.

“Strength of pathogenicity”  It is unclear what this means.  

It is unclear what the authors believe GW4869 should be used for.  In one sentence they seem to think that it is a therapeutic but then they state that it promotes biofilm growth and filamentation.  It is unclear how GW4869  murine model would lead to the development of  potential antimicrobial therapeutics.  

Reviewer 3 Report

In this manuscript, the authors investigated the roles of GW4869, a sphingomyelinase, in oral candidiasis in mice. Administrating the mice that have developed candidiasis with GW4869 for 4 consecutive days has exacerbated the fungal infections in mice in many ways. Despite many assays showed enhanced host inflammatory responses and significant tissue damage in the GW4869 group, it is too hasty to draw any firm conclusions while lacking some critical experiments.

Fungal EVs could provide host immune-protection against subsequent fungal infections. GW4869 inhibit the production and release of fungal extracellular vesicles. However, does GW4869 inhibit the formation of extracellular vesicles in host cells as well? Have the authors tested if adding the GW4869 alone would result in the keratin desquamation on the tongue mucosal surface (fig 5) and mitochondria morphological changes in epithelial cells (fig 6)?

Importantly, the manuscript is poorly written.  The gramma and word choices significantly impaired the readability of this manuscript.  This manuscript requires a thorough language and gramma polishing.

Here I list some of the places that requires improvements:

Line 12-14. It would make so much more sense if the authors could be more specific on if the EVs are from pathogens or from the host, since both pathogens and hosts both make EVs.  This should apply to the whole text.

Line 14. “inhabits” should be “inhibits” .

Line 16. “candida” should be “Candida” and “anti-candida” should be “anti-Candida”. The genus name should be capitalized and italicized. More corrections needed in line 31, 35, 46… etc

Line 30. The authors did not describe any disease or infections in the previous sentence, thus should not be using “These symptoms” as the start of the next sentence. Plus, “life-threatening systemic infections with highly mortality rates” is not a symptom!

Line 33. The authors should be specific in wording. The Candida biofilm should be one of the reasons to antifungal treatment failure, not “the reason”.

Line 126-127. Poor and unprofessional word choices. “sealed” meaning “blocked”? “Throwing away 3% BSA solution”?

Line 153. The authors claims that Figure 3A shows the biofilms were denser and greater in numbers for the GW4869 group, but the Figure 3a did not shown anything related to biofilms at all!

Line 216. “of” should be “.”

Line 220-225. The increased calcofluor white staining intensity and the increased cell wall thickness do not support the conclusion of increased Candida viability. Additionally, the fluorescent intensity of the calcofluor white staining not necessarily reflects the fungal burden: 1. It depends on the sections of the slides and the parts of the tongue. 2. It could be the upregulation of chitin on the cell wall. The authors may consider to quantify the fungal burden with the real-time PCR assay on the fungal DNA content.

Reviewer 4 Report

Well done! A very good quality of work on GW4869. Though there are a few numbers of articles related on fungal GW4869 that were published, but different angle of research needs to be addressed. There is scientific merit and overall, it will benefit all the research community for the future anti-fungal therapeutics. 

Author Response

We would like to thank you for your thorough review and favorable decision. Thank you for your insightful and helpful reviews. We are really grateful for the great efforts and much time you have spent on our manuscript.

Round 2

Reviewer 2 Report

I think that the paper could still use an editorial once over.  For instance in the discussion chin is used instead of chitin but over all my concerns over clearness of the data presented are addressed.

Author Response

Dear Reviewer,

Thank you so much for your insightful and helpful reviews. We are really grateful for the great efforts and much time you have spent on our manuscript. We have checked the manuscript and revised the typos in the revised version. Please find attached the revised manuscript with track changes. 

Sincerely,

Xiang Wang

Reviewer 3 Report

The manuscript is greatly improved after the revision, while there are still some typos (e.g. "chin" instead of "chitin" in the discussion) in the manuscripts and the not italicized Genus/Species names in the maintext (e.g. page1) and in the references.  The authors shall go through the text and make corrections.

Author Response

Dear Reviewer,

Thank you so much for your insightful and helpful reviews. We are really grateful for the great efforts and much time you have spent on our manuscript. We have checked the manuscript and revised the typos. Moreover, we have showed italicized Genus/Species names in the main text and in the references in the revised version with track changes. 

Sincerely,

Xiang Wang